# Parameter Identification of Model for Piezoelectric Actuators

**DOI:** 10.3390/mi14051050

**Published:** 2023-05-15

**Authors:** Dongmei Liu, Jingqu Dong, Shuai Guo, Li Tan, Shuyou Yu

**Affiliations:** 1College of Electronic Information Engineering, Changchun University, Changchun 130022, China; liudm@ccu.edu.cn (D.L.); 210401134@mails.ccu.edu.cn (S.G.); tanl88@ccu.edu.cn (L.T.); 2Department of Control Science & Engineering, Jilin University, Changchun 130012, China; shuyou@jlu.edu.cn

**Keywords:** piezoelectric actuators, multi-valued mapping, frequency-dependent, particle swarm genetic hybrid method

## Abstract

Piezoelectric actuators are widely used in high-precision positioning systems. The nonlinear characteristics of piezoelectric actuators, such as multi-valued mapping and frequency-dependent hysteresis, severely limit the advancement of the positioning system’s accuracy. Therefore, a particle swarm genetic hybrid parameter identification method is proposed by combining the directivity of the particle swarm optimization algorithm and the genetic random characteristics of the genetic algorithm. Thus, the global search and optimization abilities of the parameter identification approach are improved, and the problems, including the genetic algorithm’s poor local search capability and the particle swarm optimization algorithm’s ease of falling into local optimal solutions, are resolved. The nonlinear hysteretic model of piezoelectric actuators is established based on the hybrid parameter identification algorithm proposed in this paper. The output of the model of the piezoelectric actuator is in accordance with the real output obtained from the experiments, and the root mean square error is only 0.029423 μm. The experimental and simulation results show that the model of piezoelectric actuators established by the proposed identification method can describe the multi-valued mapping and frequency-dependent nonlinear hysteresis characteristics of piezoelectric actuators.

## 1. Introduction

In recent years, piezoelectric actuators (PEAs) have attracted much attention due to their advantages, such as a large output force, compact size, high resolution, and quick response. They have a wide range of potential applications in the fields of micro-manipulation [1], micromechanical manufacturing [2], super-precision processing [3], adaptive optical compensation [4,5,6], and robotics [7,8,9,10]. However, piezoelectric ceramic materials have hysteretic nonlinear characteristics of multi-valued mapping and frequency dependence [11]. The hysteretic nonlinearity prevents PEAs’ positioning precision from being further improved and impairs the stability of the closed-loop system [12].

The characteristics of frequency-dependent hysteresis and the multi-valued mapping are shown in Figure 1. The hysteresis loop between the input voltage and the output displacement reflects the multi-valued mapping [13], and the characteristic that the area of the hysteresis loop grows as the frequency of the input voltage increases reflects the frequency-dependence [14].

To improve the control accuracy of PEAs, much research has been carried out on the modeling of its hysteresis nonlinearity [15,16,17]. At present, the Hammerstein model is the most frequently used mathematical model. The Hammerstein model is a nonlinear model in which the static nonlinear section is connected in a series with the dynamic linear section [18]. The static nonlinear section can describe the multi-valued mapping properties of the hysteresis. The dynamic linear section can represent the frequency-dependent characteristics of hysteresis. Reference [4] proposes an online parameter estimating method of the Hammerstein model, which uses the Duhem model to describe the static section and the Diophantine function of the linear section to predict the output of the model. Finally, the model parameters are identified online based on the real-time updated model’s residual and cost function. Reference [19] uses the fractional order model to describe the dynamic linear section of the Hammerstein model and proposes an artificial bee colony algorithm to identify the order and coefficients of the fractional order model simultaneously.

The block structure of the Hammerstein model is convenient for expressing multi-valued mapping and frequency-dependent characteristics of hysteresis nonlinearity. The Bouc–Wen model, based on differential equations, has the advantages of fewer unknown parameters and high modeling accuracy, and it is currently widely used in the modeling and compensation of the multi-valued mapping characteristics of PEAs [20,21,22,23]. Reference [12] proposes an improved Bouc–Wen model that introduces the input voltage frequency, in which a more accurate description of the frequency-dependent hysteresis features is provided. Reference [24] proposes a generalized Bouc–Wen model, with a relaxation function for the static part of the Hammerstein model, which is used to describe the multi-valued mapping hysteresis characteristics of PEAs.

At present, the parameter identification methods of the Bouc–Wen model include the particle swarm optimization (PSO) algorithm and genetic algorithm (GA), et al. [25,26,27,28]. These algorithms have their defects, such as the PSO algorithm’s propensity to easily enter the local optimum and the GA’s poor capacity for searching for the local optimal solution [29]. These defects decrease the accuracy of the identified parameters of the Bouc–Wen model and adversely have a negative impact on the model-based control strategy. In order to solve the above problems, this paper proposes a particle swarm genetic hybrid algorithm (G-Pmix) that combines the advantages of the PSO algorithm and GA to identify the parameters of the Bouc–Wen model. The method can prevent the PSO algorithm from entering a local optimum and improve the modeling accuracy because the GA’s selection, crossover, and mutation operations are random.

A feed-forward compensator is established for the static hysteresis multi-valued mapping characteristics of the PEAs [30]. The inverse of the Bouc–Wen model as the feed-forward compensator is used to linearize the characteristics of hysteresis multi-valued mapping. The output signal will be mixed with the modeling errors, high-order unmodeled dynamics, the error caused by insufficiently compensating for the multi-valued mapping characteristics of the hysteresis, and the error from the frequency-dependent characteristics of the hysteresis only when the feed-forward controller acts on the PEAs. This paper adopts the sliding mode controller on the feedback loop and the inverse of the Bouc–Wen model as the feed-forward compensator to build a compound control system, and the sine signal tracking experiments verify the effectiveness of the model in resolving the error mentioned in the previous description and effectively suppressing chatter.

The organizational structure of this paper is as follows: Section 2 introduces the characteristics of the static section and the dynamic section of the Hammerstein model. Section 3 introduces the particle swarm genetic hybrid parameter identification algorithm. Section 4 introduces the modeling of the PEAs. Section 5 introduces the curve fitting experiments and feed-forward and feedback control experiments. Section 6 shows the conclusion of the paper.

## 2. Hammerstein Model

The multi-valued mapping and frequency-dependent hysteresis nonlinear characteristics of piezoelectric actuators severely limit the improvement of the positioning system’s accuracy. Therefore, it is necessary to establish the nonlinear hysteretic model of piezoelectric actuators to conduct research on relevant algorithms for the piezoelectric control.

The Hammerstein model is used to characterize the PEAs. The hysteresis characteristics of the PEAs are described by the static nonlinear section. The linear characteristics abstracted from the PEAs are described by the dynamic linear section. The structure block diagram is shown in Figure 2.

The Bouc–Wen model, expressed in terms of nonlinear differential equations, is used to depict the hysteresis characteristics of the PEAs and has the characteristics of fewer unknown parameters and higher model accuracy. The mathematical expression of the Bouc–Wen model is shown in Equation (1).
(1)$$\left\{\begin{array}{c} v=du-h\\ \dot{h}=\alpha d\dot{u}-\beta \left|\dot{u}\right|{\left|\dot{h}\right|}^{n-1}h-\gamma \dot{u}{\left|h\right|}^{n} \end{array}\right.$$
where u is the input voltage, v is the output displacement, h is the hysteresis of the model, $\dot{u}$ and $\dot{h}$ are the derivatives of the u and h to time, respectively, n is the smoothness of the hysteresis curve, and the value of n is 1. The size of the hysteresis loop is affected by α and d. The shape of the hysteresis loop is affected by β and γ. α,β,γ, and d are unknown parameters describing the hysteresis characteristics and need to be identified.

The mass-spring-damper system is used to describe the dynamic section of the Hammerstein model. Without considering the multi-valued mapping characteristics of hysteresis, the PEA is a mass-spring-damper linear system from the perspective of the mechanical structure [31]. The force analysis diagram is shown in Figure 3.

Where m is the mass of the PEA, y is the actual displacement, k is the elastic coefficient, c is the damping coefficient, and F is the driving force generated by the piezoelectric ceramics. According to Newton’s second law, the differential equation of the mass-spring-damper system is obtained and shown in Equation (2).
(2)$$m\ddot{y}+c\dot{y}+ky=F$$

The driving force, F, has a linear relationship with the reference input, u. By setting 

$x_{1}=y$, $x_{2}=\dot{y}$, Equation (3) is obtained.
(3)$${\dot{x}}_{2}=-\frac{k}{m}x_{1}-\frac{c}{m}x_{2}+bu$$

The model of the PEA is derived by Equations (2) and (3), as shown in Equation (4).
(4)$$\left\{\begin{array}{l} \left[\begin{array}{c} {\dot{x}}_{1}\\ {\dot{x}}_{2} \end{array}\right]=\left[\begin{array}{cc} 0 & 1\\ -\frac{k}{m} & -\frac{c}{m} \end{array}\right]\left[\begin{array}{c} x_{1}\\ x_{2} \end{array}\right]+\left[\begin{array}{c} 0\\ b \end{array}\right]u\\ y=\left[\begin{array}{cc} 1 & 0 \end{array}\right]\left[\begin{array}{c} x_{1}\\ x_{2} \end{array}\right] \end{array}\right.$$

In order to identify the parameters, the time domain model is transformed into the frequency domain model. The transfer function of the PEA is shown in Equation (5).
(5)$$G\left(s\right)=\frac{b}{s^{2}+\frac{c}{m}s+\frac{k}{m}}$$

## 3. Particle Swarm Genetic Hybrid Parameter Identification Method

According to the static and dynamic characteristics of the Hammerstein model, the hysteresis characteristics of the PEA are decomposed into multi-valued mapping and frequency-dependent characteristics. The intermediate variable, v(t), in the model, cannot be measured in the actual experimental process, so the parameter identification methods and identification sequence of the static sections and dynamic sections of the model are not unique. The step identification method is adopted to identify the static section’s parameters first and then the dynamic section’s parameters in this paper.

### 3.1. Parameter Identification of Static Nonlinear Section

In the PSO algorithm, the individual optimal position and the group optimal position are used as the principles for adjusting the velocity and position of the particles. The group optimal solution and the individual optimal solution are preserved by learning from the previous search experience to avoid blindly searching for the best solution [32,33]. The schematic diagram of the particle update is shown in Figure 4.

The GA obtains the next-generation population through crossover and mutation operations based on the competition mechanism. The whole population moves towards the global optimal solution evenly. The population can produce new and different individuals to ensure diversity [34].

By analyzing the distinct features of the PSO algorithm and GA, three conclusions are obtained as follows. 

(1) Both the PSO algorithm and GA are parallel algorithms. They all operate on multiple individuals in the population at the same time. The PSO algorithm operates on multiple particles, while the GA operates on multiple chromosomes.

(2) The evaluation principle of the solutions can be unified for the PSO algorithm and GA. They are all searching for the optimal solution in the search space based on the fitness function.

(3) It is possible to achieve unity in the coding structure for the PSO algorithm and GA. In general, the binary or real number encoding mode is adopted. Real coding maps the solution space of a problem to the real number space and operates on the real number space. This method is suitable for multi-dimensional and high-precision continuous function optimization problems and meets the requirements of this paper.

To address the weak local search ability of the GA and easily fall into a local optimal solution of the PSO algorithm, a G-Pmix parameter identification algorithm is proposed in this paper. The steps of the G-Pmix algorithm are as follows:

1. The parameters of the G-Pmix algorithm are initialized, and the population is generated randomly. Set the initial value of the position and velocity for each particle in the population and then calculate the initial fitness value of each particle. The root mean square error is selected as the fitness function, and the expression is shown in Equation (6).
(6)$$fitness(\alpha ,\beta ,\gamma ,d)=\sqrt{\frac{1}{N}\sum_{i=1}^{N}{\left(x_{exp}(i)-x_{mdl}(i)\right)}^{2}}$$
where, α,β,γ, and d are the parameters of the Bouc–Wen model to be identified, $x_{exp}(i)$ is the experimental data, $x_{mdl}(i)$ is the model data, N is the number of data sampling points, and N=6000.

Calculate the particles’ fitness values in the initial population, with which we can initialize the individual optimal position, Pbest. Search for the particle with the smallest fitness value in the initial population and take the fitness of the particle as the group optimal position, Gbest.

2. Determine whether the G-Pmix algorithm reaches the maximum number of iterations, Maxgen, or whether the group optimal position, Gbest, meets the accuracy requirement ($accurcy=1e^{-5})$. If not, optimize the population, T times (T=100), with the PSO. The group optimal position, Gbest, is updated at once after the PSO algorithm is performed every time. If either of the conditions is satisfied, quit the cycle.

The velocity and position of the particles are updated according to Equations (7) and (8).
(7)$$x_{id}(t+1)=v_{id}(t+1)+x_{id}(t)$$
(8)$$v_{id}\left(t+1\right)=\omega \left(t\right)v_{id}\left(t\right)+c_{1}\left(t\right)r_{1}\left({Pbest}_{id}\left(t\right)-x_{id}\left(t\right)\right)+c_{2}(t)r_{2}(Gbest(t)-x_{id}(t))$$
where the position and velocity information of the particle can be respectively expressed as $x_{i}=(x_{i1},x_{i2},x_{i3},x_{i4})\;1\leq i\leq S$ and $v_{i}=(v_{i1},v_{i2},v_{i3}{,v}_{i4})\;1\leq i\leq S$, S is the number of particles, $r_{1}$ and $r_{2}$ are random numbers in [0, 1], $c_{1}$ and $c_{2}$ represent the cognitive learning factor and social learning factor, and ω is the inertia weight coefficient. $c_{1}$, $c_{2}$,and ω are set as Equations (9)–(11):(9)$$c_{1}(t)=\frac{t}{T}(c_{1f}-c_{1s})+c_{1s}$$
(10)$$c_{2}(t)=\frac{t}{T}(c_{2f}-c_{2s})+c_{2s}$$
(11)$$\omega (t)=\omega_{max}-(\omega_{max}-\omega_{min})*t/T$$

3. The particles in the population are sorted by the fitness value. The mean value of the fitness value of each particle is calculated. $M_{k}$, the high-quality particles that fall below the mean value, are selected and sent to the next-generation population.

4. The crossover and mutation operations of the GA are used to optimize the remaining $S-M_{k}$ particles. The crossover operation can exchange some genes of the particles in the population with the crossover probability, ($P_{c}$), and the mutation operation can make some genes of the particles change with the mutation probability, ($P_{m}$). The crossover and mutation operations can generate new individuals and enrich the diversity of the population.

5. Combine the particles before and after optimization with the GA. The combined number is $2(S-M_{k})$. Sort the particles according to the fitness value and then single out $S-M_{k}$, the high-quality particles, and finally send them to the next generation.

6. The high-quality particles from Step 4 merge with those from Step 5, and then the next-generation population is formed.

7. Repeat steps 2–6 until the maximum number of iterations or accuracy meets the requirements, quit the cycle, and the identification is completed.

The flow chart of the G-Pmix algorithm is shown in Figure 5.

### 3.2. Parameter Identification of Dynamic Linear Section

In this paper, the System Identification Toolbox in Matlab/Simulink software was used to identify the second-order linear system. The System Identification Toolbox can construct the mathematical model based on the measured input and output data. The model parameters are identified by the least square method in the toolbox. The diagram of the dynamic linear section parameter identification is shown in Figure 6.

## 4. Modeling Based on G-Pmix Algorithm

### 4.1. Experiment Equipment

To verify the effectiveness of the particle swarm genetic hybrid algorithm, parameter identification experiments were carried out on the piezoelectric experimental platform, as shown in Figure 7. This equipment consists of an industrial computer, IPC-610L, a data acquisition card, PCI-1710, an integrated control power supply, E01, and a piezoelectric actuator, P11-X100S. The microscopic deformation of piezoelectric ceramics can be directly measured by the resistive pressure transducer strain gauge inside the piezoelectric actuator, and its measurement precision is 3 nanometers.

In order to obtain the experimental data, the control signal generated by the Matlab software is outputted through a data acquisition card and transmitted to the integrated control power supply, where the control voltage is amplified to drive the piezoelectric actuator. The corresponding output displacement of the piezoelectric ceramic is measured by the resistive pressure transducer strain gauge and transmitted back to Matlab via an integrated control power supply and data acquisition card for further processing.

### 4.2. Experiments of Static Nonlinear Section

The shape of the PEA hysteresis loop does not change when the frequency of the input signal is lower than 5 Hz [3]. Therefore, take a sine signal (f=0.5 Hz) as the input signal of the piezoelectric actuator in the parameter identification process of the static nonlinear section. The parameter setting of the G-Pmix algorithm is shown in Table 1. The fitness function curve is shown in Figure 7. The parameter identification results of the Bouc–Wen model and minimum fitness are shown in Table 2. 

Figure 8 shows that the GA can converge extremely quickly, and the convergence accuracy is the lowest due to the weak local search ability. The G-Pmix algorithm prevents the population from falling into a local optimum through the selection, crossover, and mutation operations of the GA. After 3224 iterations, the G-Pmix and PSO algorithms satisfy the convergence requirements and reach the optimal result. The minimum fitness of the G-Pmix algorithm is 6.608% lower than the PSO algorithm. The experimental results prove that the G-Pmix algorithm combines the advantages of the PSO algorithm and GA and achieves higher parameter identification accuracy.

### 4.3. Experiments of Dynamic Linear Section

Take the 0.1~50 Hz sine sweep signal, u(t), as the input to trigger the PEA experimental platform to obtain the output y(t) of the platform. The same sine sweep signal, u(t), is used as the input of the identified Bouc–Wen model and then calculate the output, v(t), of the model. Take v(t) and y(t) as the input and output data of the System Identification Toolbox. The dynamic linear section parameter identification flow chart is shown in Figure 9. The parameters of the identified second-order linear system are shown in Table 3.

## 5. Model Verification

### 5.1. Model Fitting Experiment

To verify the accuracy of the identified Hammerstein model, different input signals are used to trigger the PEA and the Hammerstein model, including a 10–50 Hz sine signal and a damped sine wave signal. Then, take the maximum fitting error (F_error) and the fitting root mean square error (F_rmse) as evaluation indicators; the solution ideas are as shown in Equations (12) and (13):(12)$$F\_\mathrm{error}=max(y_{G-Pmix/GA/PSO}-y_{Exp})$$
(13)$$F\_rmse=\sqrt{\frac{1}{N}\sum_{i=1}^{N}{\left(y_{G-Pmix/GA/PSO}\left(i\right)-y_{Exp}\left(i\right)\right)}^{2}}$$
where $y_{G-Pmix/GA/PSO}$ is the simulation output of the Hammerstein model identified by different parameter identification algorithms; $y_{Exp}$ is the actual output of the PEA.

Compare the simulation output of the Hammerstein model identified based on the GA, PSO, and G-Pmix algorithms and the actual output of the PEA. In Figure 10, the left is the data fitting curve, and the right is the fitting error curve. From Figure 10a–e, the input is a sine signal (f=10,20,30,40,50 Hz). In Figure 10f, the input is a damped sine wave signal (f=0.5 Hz). The F_error and the F_rmse are shown in Table 4. 

Figure 10 and Table 4 show that the model identified by the G-Pmix algorithm has better fitting to experimental data under the same trigger signal from 10 Hz to 50 Hz than the GA and PSO algorithm. The F_error of the G-Pmix is 25.25% lower than the PSO algorithm. The F_rmse of the G-Pmix is 24.47% lower than the PSO algorithm. 

In order to verify the accuracy of the model for multi-valued mapping and frequency-dependent characteristics, the hysteresis loop curves of different frequencies from 10 Hz to 50 Hz are plotted, as shown in Figure 11.

The models identified by the GA, PSO algorithm, and G-Pmix algorithm can characterize the multi-valued mapping and frequency-dependent nonlinear hysteresis of the PEA, as shown in Figure 11a–e. According to the result of Figure 10, it can be concluded that the model identified by the G-Pmix algorithm is more satisfied than those identified by the PSO algorithm and GA. 

### 5.2. Feed-Forward and Feedback Control Experiment

To verify the availability of the identified model, a compound control strategy with the feed-forward and feedback is proposed. Take the inverse of the Bouc–Wen model as a feed-forward controller in a series in front of the PEA to linearize the controlled object and compensate for the hysteretic nonlinear characteristics of the PEA. Then, take the superposition of the hysteresis compensation error of the feed-forward controller, high-order unmodeled dynamics, and the error between the reference input and the actual output as a total error and adopt the sliding mode controller in the feedback loop to improve the control accuracy. The structural block diagram of the compound control system is shown in Figure 12.

#### 5.2.1. Feed-Forward Control

The hysteresis characteristics of the reference input, $u_{0}$, and the compensation input, u, are opposite [11]. It is expressed as a reciprocal relationship in the mathematical expression, as shown in Figure 13. 

Therefore, the reference input, $u_{0}$, and the actual output, y, (Figure 10) have an approximately linear relationship. The mathematical expression of the Bouc-Wen inverse model is shown as Equation (14).
(14)$$\left\{\begin{array}{c} u_{0}=\frac{1}{d}(u+h)\\ \dot{h}=\alpha d\dot{u_{0}}-\beta \left|\dot{u_{0}}\right|{\left|\dot{h}\right|}^{n-1}h-\gamma \dot{u_{0}}{\left|h\right|}^{n} \end{array}\right.$$

Take the sine signal with 0.5 Hz as the input of the identified Bouc-Wen model and the feed-forward controller; the experimental results obtained are shown in Figure 14.

#### 5.2.2. Feedback Control

Figure 13 shows that the Bouc-Wen inverse model achieves the goal of linearizing the controlled object. However, the Bouc-Wen model can only characterize the static nonlinear characteristics of the hysteresis of the PEA [14]. Therefore, the feed-forward controller also needs to cooperate with the feedback controller to realize the compensation of the frequency-dependent hysteresis nonlinear characteristics of the PEA. The sliding mode control law is achieved based on the dynamic linear section of the Hammerstein model, combined with the exponential approach law to reduce the influence of chattering by reducing the parameters, ϵ, and increasing the parameters, ρ [35]. The control law of the sliding mode controller is shown in Equation (15):(15)$$u=\frac{1}{b}[\epsilon \cdot sgn\left(s\right)+\rho \cdot s+C\left(\dot{x}-x_{2}\right)+\left(\ddot{x}+\frac{k}{m}x_{1}+\frac{c}{m}x_{2}\right)]$$
where u is the control law, s is the sliding mode function, $s=C\left(x-x_{1}\right)+(\dot{x}-x_{2})$, C, ϵ, and ρ are the parameters of the sliding mode controller, C>0, ϵ>0, ρ>0, b, and c/m and k/m are the parameter identification results in Section 4.2.

#### 5.2.3. Signal Tracking Experiment

The parameter settings of the sliding mode controller are shown in Table 5. The reference inputs are (a) the ramp signal, (b) the damped sine wave signal, (c) the sine signal with 10 Hz, (d) the sine signal with 30 Hz, (e) the sine signal with 50 Hz, and (f) the complex frequency signal with 10~50 Hz.

The experimental results are shown in Figure 15 by comparing the model identified by the G-Pmix with the PSO algorithm. The left figure is the curve of the reference input and the actual output, and the right figure is the tracking error curve. The maximum tracking error (T_error) and the tracking root mean square error (T_rmse) are shown in Table 6.

Table 6 shows that both the PSO algorithm and G-Pmix provide great control performance, while the ramp signal is used as the input signal, and the T_error of the G-Pmix is less than that of the PSO algorithm. Due to the fixed frequency and gradient of the ramp signal, T_rmse, obtained by the G-Pmix, is equal to T_rmse, obtained by the PSO algorithm. The advantages of the G-Pmix are it grows as the complexity of the input signals rises. The availability of the G-Pmix algorithm proposed in this paper for identifying model parameters is once more verified.

## 6. Conclusions

This paper adopts the Hammerstein model to describe the multi-valued mapping and frequency-dependent characteristics of PEAs. The static multi-valued mapping is characterized by the Bouc–Wen model, while the dynamic frequency-dependent characteristic is described by the mass-spring-damper model. To identify the parameters of the Bouc–Wen model, the G-Pmix algorithm is proposed in the paper, which combines the strong local optimization ability of the PSO algorithm and the strong global optimization ability of the GA. The population diversity of the PSO algorithm is enriched by introducing the random characteristics of the selection, crossover, and mutation processes of the GA, and the combination of both algorithms can improve the accuracy of parameter identification. The Bouc–Wen model parameter identification experiments show that the minimum fitness of the G-Pmix is 0.029423 μm, which is 6.608% lower than that of the PSO algorithm. According to experimental results above, the G-Pmix has a greater capability for optimization than the PSO algorithm. The parameters of the linear section of the Hammerstein model are identified with the System Identification Toolbox. The experimental results of the parameter identification demonstrate that the Hammerstein model, established by the G-Pmix, can characterize the multi-valued mapping and frequency-dependent nonlinear hysteretic of the PEA, and the output of the Hammerstein model can match the real output. The feed-forward inverse model controller and the feedback sliding mode controller are established based on the Hammerstein model. Signal tracking experiments show that the T_error and T_rmse will rise with the increased complexity of the input signal when the chattering phenomenon is restrained. Signal tracking experimental results prove that the model parameters identified by the G-Pmix can make the model-based feed-forward and feedback controllers achieve superior control effects and enable the PEA to obtain greater control precision than the PSO algorithm.

## Figures and Tables

**Figure 1 micromachines-14-01050-f001:**
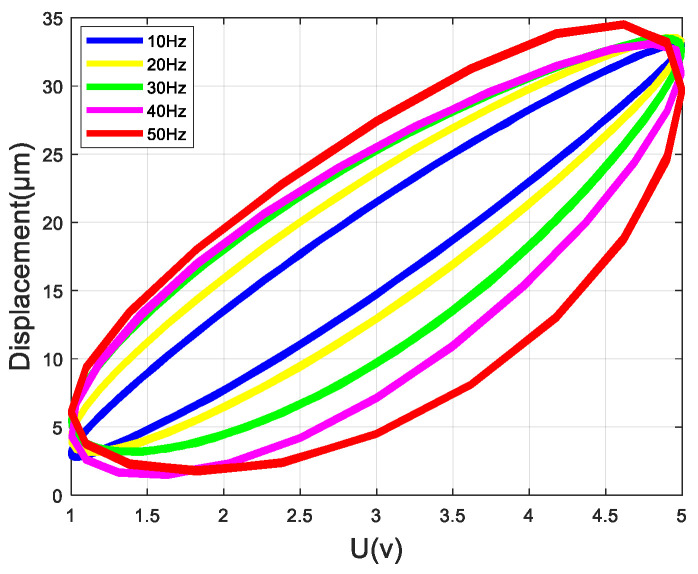
Multi-valued mapping and frequency-dependent hysteresis curve of PEAs.

**Figure 2 micromachines-14-01050-f002:**
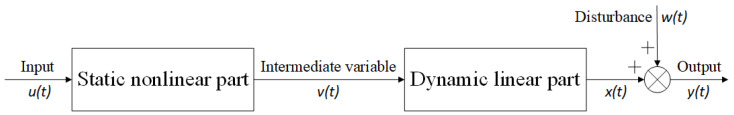
Structure block diagram of the Hammerstein model.

**Figure 3 micromachines-14-01050-f003:**
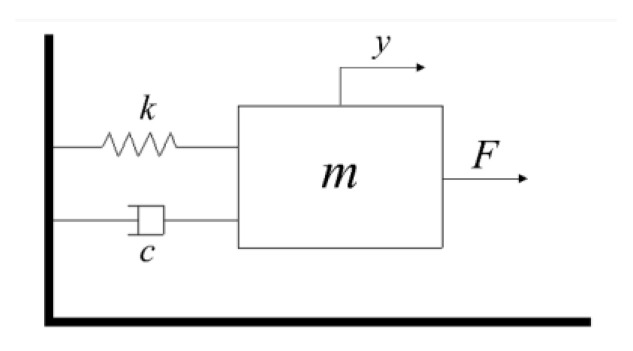
Force analysis diagram of the PEA.

**Figure 4 micromachines-14-01050-f004:**
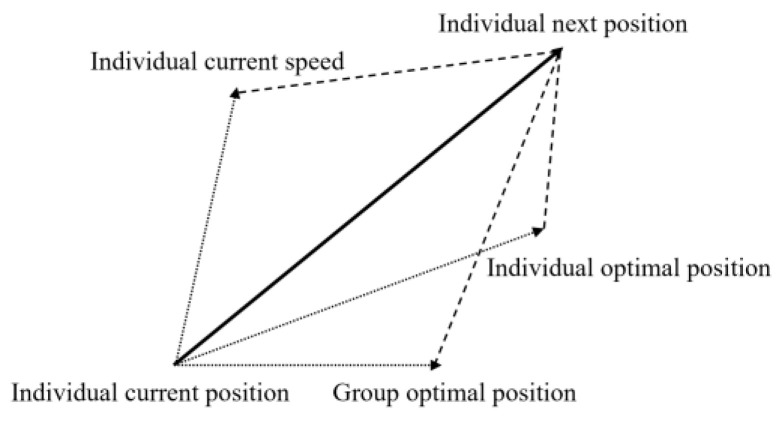
Schematic diagram of the particle update.

**Figure 5 micromachines-14-01050-f005:**
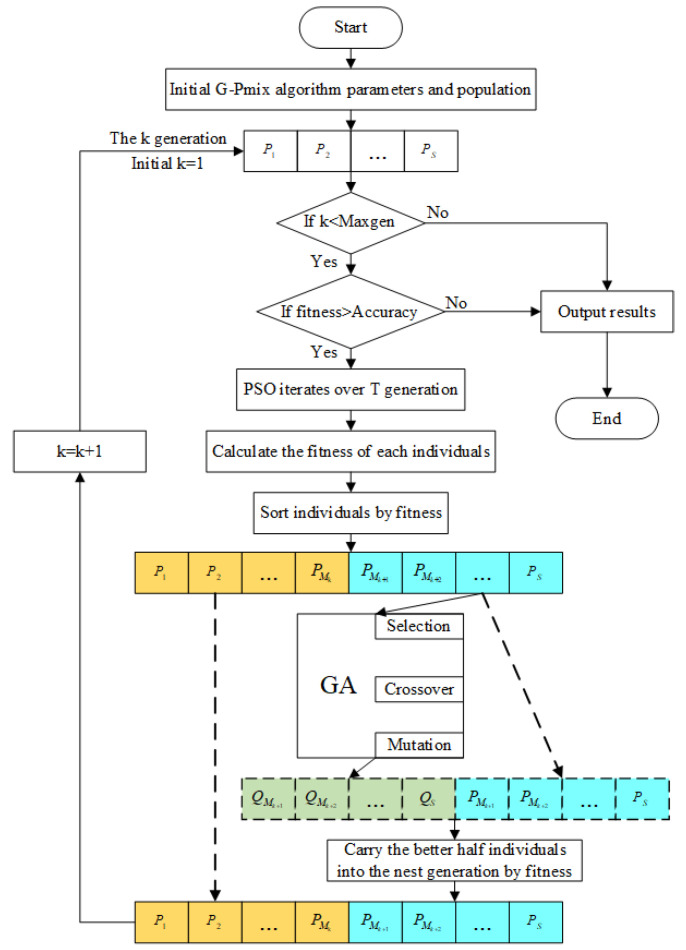
Flow chart of the G-Pmix algorithm.

**Figure 6 micromachines-14-01050-f006:**
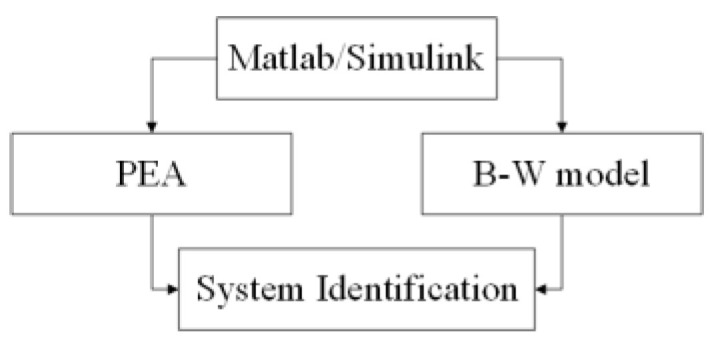
Parameter identification diagram of the dynamic linear section.

**Figure 7 micromachines-14-01050-f007:**
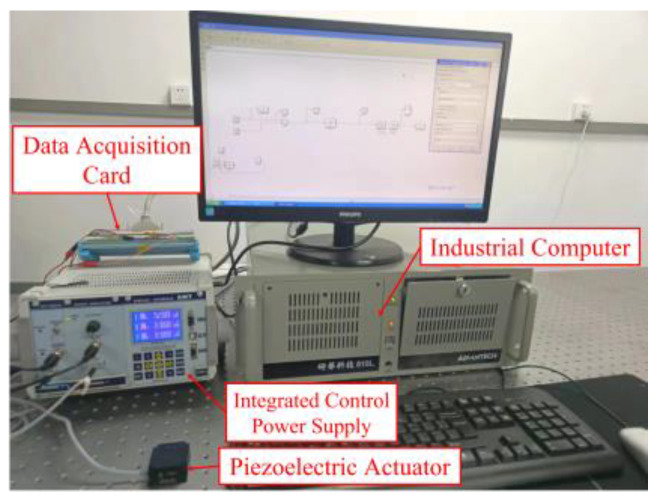
Experimental equipment.

**Figure 8 micromachines-14-01050-f008:**
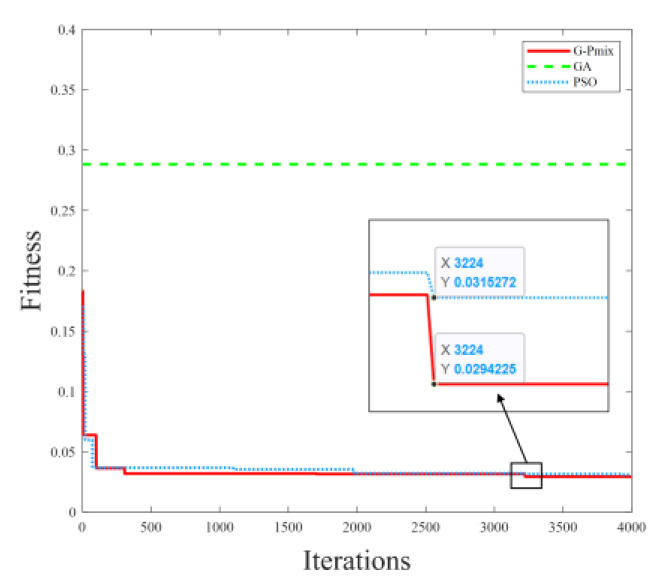
Fitness function curve of the Bouc–Wen model’s parameter identification experiment.

**Figure 9 micromachines-14-01050-f009:**
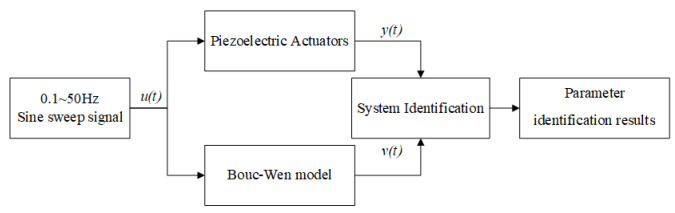
Flow chart of dynamic linear section parameter identification.

**Figure 10 micromachines-14-01050-f010:**
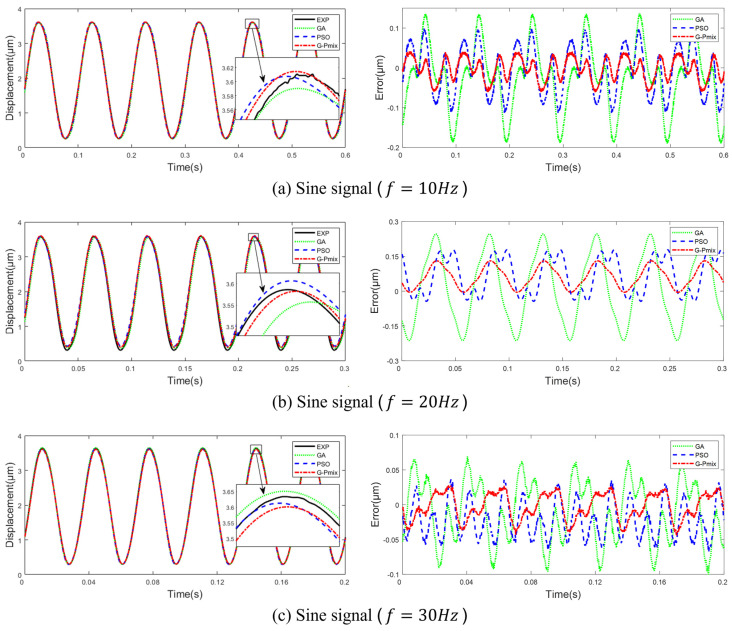
Data fitting curve and fitting error curve.

**Figure 11 micromachines-14-01050-f011:**
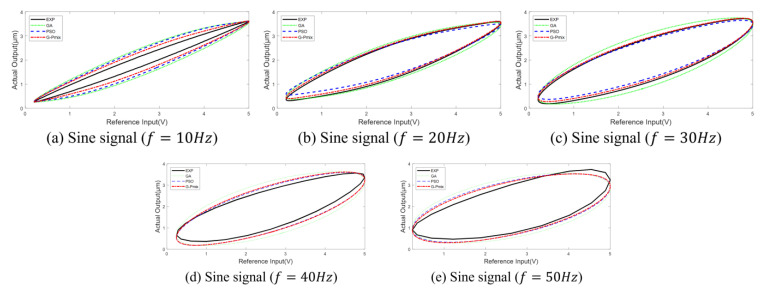
Hysteresis loop curve.

**Figure 12 micromachines-14-01050-f012:**
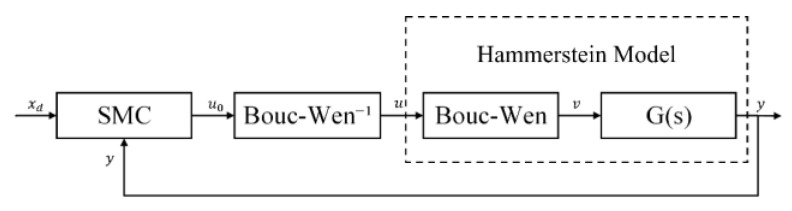
Structural block diagram of the compound control system.

**Figure 13 micromachines-14-01050-f013:**
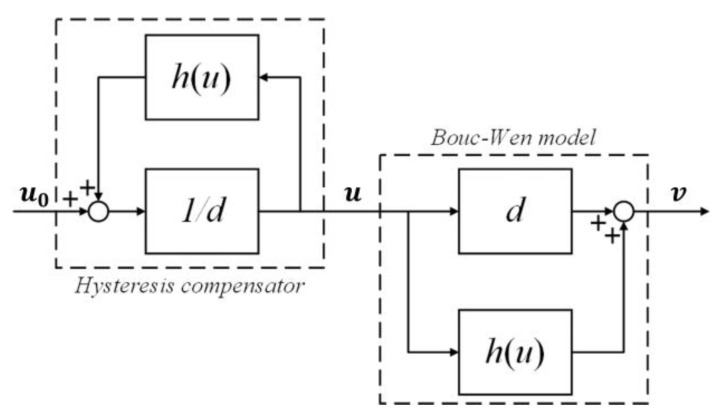
Structure block diagram of the feed-forward control principle.

**Figure 14 micromachines-14-01050-f014:**
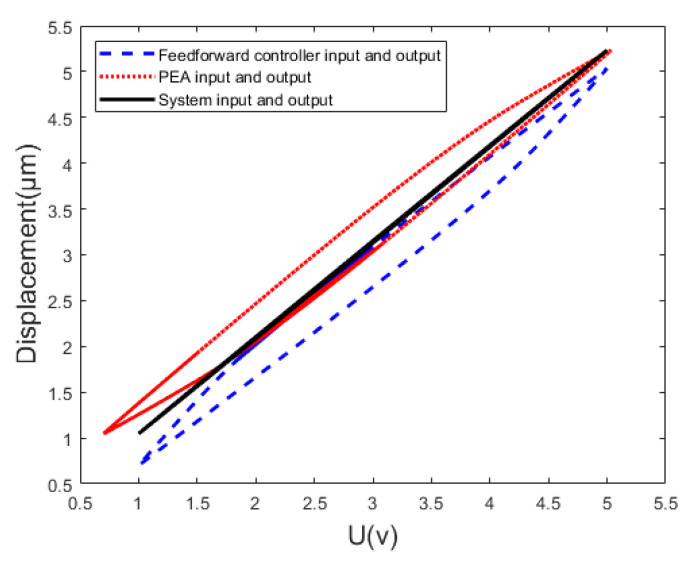
Feed-forward control simulation results.

**Figure 15 micromachines-14-01050-f015:**
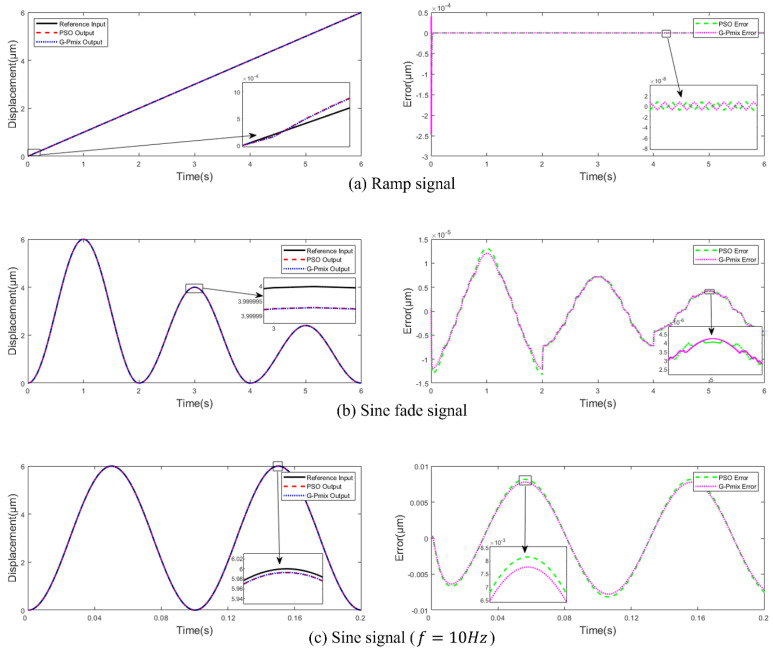
The experiment results of the compound control with the feed-forward and feedback: (**a**) ramp, (**b**) damped sine wave, (**c**) sine 10 Hz, (**d**) sine 30 Hz, (**e**) sine 50 Hz, and (**f**) complex frequency.

**Table 1 micromachines-14-01050-t001:** Parameter setting list of the G-Pmix algorithm.

Parameter	Maxgen	S	$c_{1s}$	$c_{1f}$	$c_{2s}$	$c_{2f}$	$\omega_{min}$	$\omega_{max}$	$P_{c}$	$P_{m}$
Value	4000	50	4	1	1	4	0.4	0.9	0.2	0.2

**Table 2 micromachines-14-01050-t002:** Bouc–Wen model’s parameter identification results.

Parameter	G-Pmix	PSO	GA
α	0.304225	0.262017	0.132514
β	0.675313	0.675520	0.409838
γ	0.497724	0.368105	1.048550
d	1.045814	1.048355	0.947412
Minimum fitness	0.029423	0.031505	0.288188

**Table 3 micromachines-14-01050-t003:** Parameter identification results of the second-order linear system.

Parameter	G-Pmix	GA	PSO
b	267,700	327,100	228,900
c/m	510.1069	683.914	386.7
k/m	255,700	284,790	225,200

**Table 4 micromachines-14-01050-t004:** F_error and F_rmse of the model fitting experiment.

Signal	G-Pmix F_error/F_rmse (μm)	PSO F_error/F_rmse (μm)	GA F_error/F_rmse (μm)
10 Hz	0.0617/0.0284	0.1134/0.0601	0.1910/0.0865
20 Hz	0.1306/0.0775	0.1781/0.1047	0.2463/0.1531
30 Hz	0.0434/0.0192	0.0688/0.0345	0.1000/0.0457
40 Hz	0.0336/0.0141	0.0706/0.0413	0.0791/0.0515
50 Hz	0.1547/0.0232	0.0684/0.0399	0.1547/0.1056
Damped sine wave	0.1784/0.1501	0.2301/0.1868	0.2451/0.2553
Mean of F_error/F_rmse	0.0823/0.0519	0.1216/0.0779	0.1694/0.1163

**Table 5 micromachines-14-01050-t005:** The parameters of the sliding mode controller.

Parameters	C	ϵ	ρ	d
Value	0.304225	0.675313	0.497724	1.045814

**Table 6 micromachines-14-01050-t006:** T_error and T_rmse of the signal tracking experiment.

Signal	G-Pmix T_error (μm)	G-Pmix T_rmse (μm)	PSO T_error (μm)	PSO T_rmse (μm)
Ramp	2.4739 × 10^−4^	0.0011	2.5372 × 10^−4^	0.0011
Damped sine wave	1.2311 × 10^−5^	9.8599 × 10^−4^	1.3309 × 10^−5^	0.0010
Sine 10 Hz	0.0078	0.0053	0.0081	0.0055
Sine 30 Hz	0.0539	0.0368	0.0564	0.0385
Sine 50 Hz	0.1134	0.0763	0.1187	0.0798
Complex frequency	0.3375	0.0910	0.3512	0.0948

## Data Availability

Data cannot be shared publicly, because data from this study may contain potentially or sensitive information. Data from this study will be made available for researchers who meet criteria for access to confidential data. Requests may be sent to: dongjingqu@foxmail.com.

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
