# Peer review of "Parameter Identification of Model for Piezoelectric Actuators"

_micromachines, 2023, doi:10.3390/mi14051050_

Round 1

Reviewer 1 Report

Dear Authors,

The manuscript should be revised before it is published. My review is attached below;

Kind Regards

Reviewer 2 Report

This work is supported by a thorough investigation of theory and experiment. The presentation of the problem and the experiment design should be improved. Figures, such as schematic or photos, can be added to explain the existing challenge, and the experimental or application setup of this study. The current presentation using mainly data curves renders a poor readibility. I believe the reviewers and future readers are all intimidated by the current presentation. 

Reviewer 3 Report

This paper addresses solving the hysteretic nonlinear effects of the piezoelectric actuators using the combination of Particle Swarm Optimization (PSO) and Genetic Algorithm (GA). In this work, the parameters of the Bouc-Wen model and the mass-spring-damper model were optimized in order to fit experimental data. The G-Pmix method, which incorporates PSO and GA, demonstrated lower error rates when compared to the individual use of either PSO or GA. The paper is clearly written, it lacks important algorithmic details and information regarding the experimental configuration. Recommend accepting it after a minor revision.

A few issues should be properly addressed before the acceptance of this paper.

1.     How many iterations does the PSO algorithm take to converge and reach an optimized result?

2.     In line 104, h should be defined before h' is defined.

3.     In line 179, the definition of ‘S’ is unclear, is it the number of particles? Because in line 186, the GA algorithm uses the remaining N-Mk particles, however, in figure 5, it's S-Mk that were used by the GA algorithm. 

4.     It would be helpful to include information on the type of actuator and sensor used in the experimental platform.

5.     It is suggested that additional references be included in the paper to highlight the application background in robotics and actuators:

[1] J. Zhao, G. Mu, H. Dong, T. Sun and K. T. V. Grattan, "Study of the Velocity and Direction of Piezoelectric Robot Driven by Traveling Waves," in IEEE Transactions on Industrial Electronics, vol. 70, no. 9, pp. 9260-9269, Sept. 2023, doi: 10.1109/TIE.2022.3210545.

[2] G. Mu, J. Zhao, H. Dong, J. Wu, K. T. V. Grattan and T. Sun, "Structural parameter study of dual transducers-type ultrasonic levitation-based transportation system," in Smart Materials and Structures 30, no. 4 (2021): 045009.

[3] J. Wu et al., "Development of a Self-Moving Ultrasonic Actuator with High Carrying/Towing Capability Driven by Longitudinal Traveling Wave," IEEE-ASME Trans. Mechatron., 2022.

[4] J. Deng, Y. Liu, J. Li, S. Zhang and K. Li, "Displacement Linearity Improving Method of Stepping Piezoelectric Platform Based on Leg Wagging Mechanism," IEEE Trans. Ind. Electron., vol. 69, no. 6, pp. 6429-6432, June 2022.

Additionally, the following typos and formatting issues should be addressed: 

1.     In Figure 4, the "individual nest position", is it 'best solution'?

2.     Clarify the subtitle of Figure 9 to improve readability. 

3.     The format of the reference section should be revised carefully.

Round 2

Reviewer 1 Report

Dear Authors,

You have revised the manuscript according to my remarks. I do not have any further comments. The manuscript can be published after minor revision (methodological errors and text editing). 

Kind Regards

Reviewer 2 Report

The research topic is a scientific problem requiring study. This work combines analytics, computation and experiments, which supports the conclusion. It is useful for the modeling and control of piezoelectric actuators.